# Prokaryotic Diversity of the Composting Thermophilic Phase: The Case of Ground Coffee Compost

**DOI:** 10.3390/microorganisms9020218

**Published:** 2021-01-21

**Authors:** Maria Papale, Ida Romano, Ilaria Finore, Angelina Lo Giudice, Alessandro Piccolo, Silvana Cangemi, Vincenzo Di Meo, Barbara Nicolaus, Annarita Poli

**Affiliations:** 1Institute of Polar Sciences, National Research Council of Italy, Spianata San Raineri 86, 98122 Messina, Sicilia, Italy; maria.papale@isp.cnr.it (M.P.); angelina.logiudice@cnr.it (A.L.G.); 2Institute of Biomolecular Chemistry, National Research Council of Italy, Via Campi Flegrei 34, 80078 Pozzuoli, Naples, Italy; iromano@icb.cnr.it (I.R.); ilaria.finore@icb.cnr.it (I.F.); bnicolaus@icb.cnr.it (B.N.); 3Centro Interdipartimentale di Ricerca sulla Risonanza Magnetica Nucleare per l’Ambiente, l’Agro-alimentare ed i Nuovi Materiali (CERMANU), Università di Napoli Federico II, Via Università 100, 80055 Portici, Naples, Italy; alessandro.piccolo@unina.it (A.P.); silvanacangemi.sc@gmail.com (S.C.); 4Dipartimento di Agraria, Università Federico II, Via Università 100, 80055 Portici, Naples, Italy; vincenzo.dimeo@unina.it

**Keywords:** coffee compost, metagenomic analysis, culture-dependent approaches, thermophiles

## Abstract

Waste biomass coming from a local coffee company, which supplied burnt ground coffee after an incorrect roasting process, was employed as a starting material in the composting plant of the Experimental Station of the University of Naples Federico II at Castel Volturno (CE). The direct molecular characterization of compost using ^13^C-NMR spectra, which was acquired through cross-polarization magic-angle spinning, showed a hydrophobicity index of 2.7% and an alkyl/hydroxyalkyl index of 0.7%. Compost samples that were collected during the early “active thermophilic phase” (when the composting temperature was 63 °C) were analyzed for the prokaryotic community composition and activities. Two complementary approaches, i.e., genomic and predictive metabolic analysis of the 16S rRNA V3–V4 amplicon and culture-dependent analysis, were combined to identify the main microbial factors that characterized the composting process. The whole microbial community was dominated by Firmicutes. The predictive analysis of the metabolic functionality of the community highlighted the potential degradation of peptidoglycan and the ability of metal chelation, with both functions being extremely useful for the revitalization and fertilization of agricultural soils. Finally, three biotechnologically relevant Firmicutes members, i.e., *Geobacillus thermodenitrificans* subsp. *calidus*, *Aeribacillus pallidus*, and *Ureibacillus terrenus* (strains CAF1, CAF2, and CAF5, respectively) were isolated from the “active thermophilic phase” of the coffee composting. All strains were thermophiles growing at the optimal temperature of 60 °C. Our findings contribute to the current knowledge on thermophilic composting microbiology and valorize burnt ground coffee as waste material with biotechnological potentialities.

## 1. Introduction

Composting organic wastes represents an important pathway for carbon flow and nutrient cycling in both developed and developing countries. Composting, often described as nature’s way of recycling, is a self-heating, aerobic, solid-phase process, during which organic waste materials are biologically degraded into an extremely useful humus-like substance. The product resulting from this process is called “compost” (from Latin compositum meaning consisting of more than one substance), which stabilizes biologically numerous types of organic waste by converting them into a final product containing a proportion of humus. The compost, which is rich in nutrients and hygienically safe, is created by reproducing processes that are found in nature that ensure the recycling of nutrients in a controlled and accelerated way. The main protagonists of this dynamic process are microorganisms (via the rapid growth of bacteria, fungi, and actinomycetes) and their enzymes [1]. Human control of the biological decomposition process is what differentiates composting from the natural decomposition of organic matter; in fact, regulating and optimizing the conditions ensures a faster process and the generation of quality end products.

The composting process is characterized by four phases: (1) the initial mesophilic phase (10–42 °C), during which, the temperature rapidly rises and initiates organic matter decomposition; (2) the thermophilic phase (45–70 °C), which is distinguished by prolonged high temperatures due to the extensive metabolic activities undertaken by endogenous microorganisms; (3) the middle mesophilic phase (65–50 °C), during which, the temperature decreases, allowing for re-establishment of the heat-resistant microbes; (4) the finishing phase (50–23 °C), during which, the organic matter and biological heat production stabilize [2]. Moreover, it is carried out by different classes of microbes, such as mesophiles and thermophiles. Generally, mesophilic microorganisms, which function best between 30 and 50 °C, initiate the composting process. As microbial activity increases soon after compost piles are formed, temperatures and density within the piles also increase and thermophilic microorganisms take over at temperatures above 50 °C. The temperature in the compost pile typically increases rapidly from 50 to 70 °C within 24 to 72 h of the pile formation and can stay there for several days depending on feedstock properties, pile size, and environmental conditions. This represents the “active phase” of composting, during which, decomposition is the most rapid. It continues until the materials containing nutrients and energy within the piles have been transformed. As the microbial activity decreases, the pile compost temperature gradually declines to approximately 37 °C. Mesophilic microorganisms recolonize the pile, and the compost enters the “curing phase.” The oxygen consumption during curing declines and organic materials continue to decompose and are converted to biologically stable humic substances that represent the mature or finished compost. Potentially toxic organic acids and resistant compounds are also stabilized during curing. A long curing phase is needed if the compost is unfinished or immature, which is possible if the compost pile contained too little oxygen or either too little or too much moisture [2] (Figure 1).

The functional roles of bacteria have been classically assessed by using culture-dependent techniques, highlighting the secretion of enzymes that are able to degrade, for example, lignocellulose during the composting process [3]. However, the relationships between microbes, their metabolism, and the biodegradation of organic material remain under-investigated. To address this issue, culturable and unculturable microorganisms need to be considered [3]. Recently, the Phylogenetic Investigation of Communities by Reconstruction of Unobserved States (PICRUSt) method was employed to predict the metabolic functions of the microbial community based on sequence data derived from high-throughput sequencing [4]. To date, only a few studies have applied the PICRUSt approach to investigate the microbial metabolism in aerobic composting [3,5,6].

The dynamics of microbiota during waste composting are strictly linked to the environmental conditions inside the compost, such as nutrients availability, starting materials, temperature fluctuations, oxygen concentration, pH, and particle size [7]. A wide variety of microbial populations (bacteria and fungi) have been associated with compost by applying culture-dependent and -independent techniques [8]. The latter have allowed for the identification of unknown sequences that are potentially ascribable to new taxa, thus unraveling a huge variety of mesophilic and thermophilic microorganisms that inhabit a compost’s multifaceted environment [8,9,10,11]. In particular, the bacterial species that dominate the thermophilic phase of compost processes generally belong to the genus *Bacillus*. They play an important role in the degradation of complex substrates, such as the cellulose polymer [12]. Recently, numerous thermophilic bacteria that are present in several composting processes were isolated and characterized, such as *Geobacillus toebii* subsp. *decanicus* [13], *Geobacillus galactosidasius* [14], and *Aeribacillus composti* [15].

In the present study, a compost sample collected during the early “active thermophilic phase” of a ground coffee composting process was microbiologically analyzed. To the best of our knowledge, for the first time, the starting material employed for composting was ground coffee obtained after an incorrect roasting procedure. Through the combined application of culture-dependent and -independent methods, this work was aimed at valorizing burnt ground coffee (i) by exploring the prokaryotic diversity (via 16S rRNA gene amplicon sequencing) and predictive metabolic potentialities (via the predictive functional analysis on 16S rRNA gene data) to identify the main microbial factors that could contribute to the evaluation and improvement of composting processes, and (ii) as a source of new bacterial isolates with high biotechnological potential.

## 2. Materials and Methods

### 2.1. Sampling of the Compost

Compost was produced in the composting plant of the Experimental Station of the University of Naples Federico II at Castel Volturno (CE); it was obtained by mixing ground coffee, fresh grass, and mature compost as a starter, approximately in the ratio of 18:80:2, respectively. The ground coffee was derived from a local coffee company that supplied the coffee material after an incorrect roasting process. The compost was obtained via an on-farm composting process based on an active or thermophilic phase that was reached with forced aeration of static piles and was followed by a two-month curing period. In the composting system, mechanical aeration was temperature-controlled and assured using air injection through a basal net of tubes connected to a blower. Pile wetting was done through an irrigation system, which was manually activated when the gravimetrically determined relative humidity (RH) was <50%. During the early stage of the thermophilic phase, pile heating exceeded 55 °C for at least 5 days to achieve sanitation. A compost sample was aseptically collected on the sixth day of composting when the temperature, as measured by PT100 thermo-sensors (Thermo Scientific, Waltham, MA, USA) placed in the core of the pile, was 63 °C.

### 2.2. NMR Spectroscopy of the Compost

A 300 MHz Bruker Avance spectrometer (Bruker Italia, Milan, Italy), equipped with a 4 mm wide-bore MAS probe, was used for solid-state analyses of the compost. A fine-powdered compost sample (250 mg) was packed into a 4 mm zirconium rotor, which was provided with a Kel-F cap, and spun at a rate of 13,000 ± 1 Hz. The ^13^C-NMR spectrum was acquired through a cross-polarization magic-angle spinning (CPMAS) technique by using 2 s of recycling delay, 1 ms of contact time, 30 ms of acquisition time, and 4000 scans. Chemical shift regions were assigned according to Cozzolino et al. [16]. The area of each region was determined by integration using the MestreNova 6.2.0 software (Mestrelab Research, S.L., Santiago de Compostela, Spain) and expressed as a percentage of the total area.

### 2.3. Isolation and Characterization of the Thermophilic Strains: A Culture-Dependent Approach

#### 2.3.1. Isolation of the Thermophilic Strains

For the isolation of the thermophilic strains, 1 g of coffee was suspended in 100 mL of ringer solution (RS), a sterile buffer saline solution at pH 7.0, composed of 0.150 g of potassium chloride, 2.25 g of sodium chloride, 0.05 g of sodium bicarbonate, and 0.12 g of calcium chloride hexahydrated per liter of distilled water and placed at 50 °C and 120 rpm for 3 h. Subsequently, 100 μL of this solution was suitably diluted and plated on the enrichment growth media Nutrient Broth (NB) (Oxoid S.P.A, Rodano, Milan, Italy) and Yeast-NaCl broth (YN) (6.0 g/L of yeast extract and 3.0 g/L of NaCl at pH 6.5 in tap water) and incubated aerobically at both 60 and 70 °C. Agar plates were prepared by adding 1.8% (*w*/*v*) of bacteriological agar (Oxoid S.P.A, Rodano, Milan, Italy) to the liquid medium.

After 48 h of incubation, several cream and white colonies had developed at the incubation temperature of 60 °C. Strains CAF2 and CAF5 were the predominant microorganisms in the NB medium, while a strain named CAF1 was the predominant microorganism in the YN medium. They were purified using the repeated serial dilution technique followed by re-streaking on solid enrichment NB and YN media. The purity of the isolates was examined based on the cell shape under phase-contrast microscopy (Nikon Eclipse E400, Nikon Europe, Badhoevedorp, The Netherlands) and colony homogeneity on the plates under a stereomicroscope (M8; Leica, Leica Microsystems, Mannheim, Germany).

#### 2.3.2. Determination Optimal Growth Conditions and Phenotypic Characterization

The optimal temperature range for the growth was tested by incubating the isolates in a liquid NB or YN medium in a temperature range from 50 to 70 °C. Bacterial growth was directly monitored in a UV/Vis spectrophotometer DU 730 (Beckman Coulter s.r.l., Milan, Italy) by measuring the absorbance at λ = 540 nm. Gram reactions were performed by assaying aminopeptidase activity using Bactident (Merck, Roma, Italy) and using the KOH lysis method [15]. The oxidase activity was determined by assessing the oxidation of tetramethyl-*p*-phenylenediamine and the catalase activity was determined by assessing the bubble production in a 3% (*v*/*v*) hydrogen peroxide solution [17]. The starch hydrolysis was tested by flooding cultures with Lugol’s iodine solution on a solid medium containing 0.2% (*w*/*v*) starch. Xylan and cellulose hydrolysis were tested by flooding cultures with 0.1% Congo Red dye, followed by rinsing with a 1.0 M NaCl solution on a solid medium containing 0.2% (*w*/*v*) xylan and carboxymethyl cellulose (CMC), respectively. Gelatine hydrolysis was tested in the appropriate medium enriched with 1.0% (*w*/*v*) gelatine [15,18]. Detection of enzymatic activities in the cytosol fraction of three isolates was performed by suspending their cells in a 50 mM Tris-HCl buffer pH 7.0 containing lysozyme (3 mg of lysozyme per gram of wet cell) at 37 °C for 1 h. The cytosolic samples obtained through centrifugation at 10,000 rpm for 30 min were tested for endoxylanase, β-xylosidase, and α-arabinofuranosidase, as previously reported [15,16,17,18,19].

#### 2.3.3. Molecular Characterization and Phylogenetic Analysis

For the DNA analysis, the cells of strains CAF2, CAF5, and CAF1 were collected after 24 h of incubation at a temperature of 60 °C via centrifugation at 5000× *g* for 15 min. Genomic DNA was isolated using DNAzol (Molecular Research Center, Inc., Cincinnati, OH, USA) according to the manufacturer’s instructions. Polymerase chain reaction (PCR) amplification and sequencing of the almost full-length 16S rRNA genes of three strains were obtained from the genomic DNA amplification by using universal primers 8F and 1517R with a broad specificity into a PCR. The nucleotide sequences of 16S rRNA genes were analyzed using the EzTaxon-e server (https://www.ezbiocloud.net) and the values for the pairwise 16S rRNA gene sequence similarities between the closest species were determined using the EzTaxon-e server. The phylogenetic tree was reconstructed by using the software package MEGA X (version 10.1 https://www.megasoftware.net/show_eua) after multiple alignments of the data using CLUSTAL_X [20]. The evolutionary history was inferred by using the maximum likelihood method and the Tamura 3-parameter model [21]. The percentage of trees in which the associated taxa clustered together is shown next to the branches. The initial tree(s) for the heuristic search were obtained automatically by applying the neighbor-joining and BioNJ algorithms to a matrix of pairwise distances that were estimated using the maximum composite likelihood (MCL) approach and then selecting the topology with the superior log-likelihood value.

### 2.4. Metagenomic Analysis: DNA Extraction from the Compost Samples

The coffee compost sample (1.5 g), which was reduced to a pulp thanks to the use of sterile pestles, was used for DNA extraction by employing the PowerSoil DNA extraction kit (MoBio Laboratories, Carlsbad, CA, USA) according to the manufacturer’s instructions. The DNA concentrations and purity were quantified by using a NanoDrop ND-1000 UV-Vis Spectrophotometer (NanoDrop Technologies, Wilmington, DE, USA). From the compost sample, three replicates (named caffA, caffB, and caffE) of the DNA extraction were created. In particular, caffE was extracted from the darkest and most compact part of the compost sample.

#### 2.4.1. Amplification of 16S rRNA Genes and Illumina Sequencing

Our target was the V3–V4 hypervariable region of the bacterial 16S rRNA gene. PCR was started immediately after the DNA was extracted. The 16S rRNA V3–V4 amplicon was amplified using two universal bacterial 16S rRNA gene amplicon PCR primers (Polyacrylamide gel electrophoresis (PAGE) purified), namely, the amplicon PCR forward primer 338F (5′-CTACGGGNGGCWGCAG-3′) and the amplicon PCR reverse primer 806R (5′-GACTACHVGGGTATCTAATCC-3′). The Illumina overhang adapter sequences that were added to the locus-specific sequences were the forward overhang 5′-TCGTCGGCAGCGTCAGATGTGT-ATAAGAGACAG-3′ and the reverse overhang 5′-GTCTCGTGGGCTCGGAGATGTGT-ATAAGAGACAG-3′.

The reaction was set up as follows: microbial DNA (5 ng/μL) 2.5 μL; amplicon PCR forward primer (1 μM) 5 μL; amplicon PCR reverse primer (1 μM) 5 μL; 2× KAPA HiFi Hot Start Ready Mix (Roche Sequencing Solution, Milan, Italy) 12.5 μL (total 25 μL). The plate was sealed and the Polymerase Chain Reaction (PCR) performed in a thermal instrument (Applied Biosystems 9700, Monza, Italy) using the following program: 1 cycle of denaturing at 95 °C for 3 min, followed by 25 cycles of denaturing at 95 °C for 30 s, annealing at 55 °C for 30 s, elongation at 72 °C for 30 s, and a final extension at 72 °C for 5 min. The PCR products were checked using electrophoresis in 1% (*w*/*v*) agarose gels in a TBE buffer (Tris, boric acid, EDTA) that was stained with ethidium bromide (EB) and visualized under UV light. A bioanalyzer (Agilent 2100, Milan, Italy) with a DNA 1000 chip was used to verify the size of the PCR product (the preparation was performed according to the “Preparing 16S Ribosomal RNA Gene Amplicons for the Illumina MiSeq System” recommendations).

AMPure XP beads (Sigma-Aldrich, Milan, Italy) were used to purify the free primers and primer dimer species in the amplicon product. To sequence the amplicon, dual indices and Illumina sequencing adapters [22] were attached using the Nextera XT Index Kit and the amplicon was purified again using AMPure XP beads. Before sequencing, the DNA concentration of each PCR product was determined using a Qubit^®^ 2.0 Green double-stranded DNA assay, which was quality controlled using a bioanalyzer (Agilent 2100, Milan, Italy). Depending on the coverage needs, all libraries could be pooled for one run. The amplicons from each reaction mixture were pooled in equimolar ratios based on their concentrations. Sequencing was performed using the Illumina MiSeq system (Illumina MiSeq, San Diego, CA, USA), according to the manufacturer’s instructions.

#### 2.4.2. Post-Run Analysis

The raw data were analyzed using the bioinformatics analysis software MOTHUR (version 1.39.5). Barcodes and primers were identified with a maximum one-base error and trimmed off. Reads were cleaned by length (reads shorter than 200 bp were discarded) and by quality score using score quality windows (i.e., average 30 and size 10). The remaining sequences were aligned with the Silva reference files (release 132 full-length sequences and taxonomy references). Reads were denoised using the pre-cluster command [23] to remove sequences that were likely due to pyrosequencing errors and assemble reads that differed only by 2 bp. Chimeric sequences were detected using the UCHIME method and removed [24]. All quality-checked sequences were taxonomically classified using the reference alignment with 100 iterations and a minimum bootstrap confidence score of 80% and were clustered into operational taxonomic units (OTUs) at a 97% similarity level. Consensus sequences of each OTU were again classified with a minimum consensus confidence threshold of 80% [25].

#### 2.4.3. Analysis of the Prokaryotic Community

To generate taxonomic profiles, the sequences were classified against the Silva database [26] and distance matrices (label 0.03) were created to generate the OTU table for subsequent analyses. Starting from these data, the matrix and rarefaction curves for both brines were calculated for a 0.03 distance cut-off [27].

The obtained OTUs for Bacteria and Archaea using 16S rRNA gene amplicon sequencing (reads were normalized to the lowest number of reads by the mother sub-sample command). Diversity index calculations were performed after subsampling based on the lowest number of reads. Several alpha diversity measures were evaluated by MOTHUR software, and the final OTU table was exported in the biom format.

#### 2.4.4. Predictive Functional Profiling

PICRUSt2 is a functional profiling prediction tool that uses evolutionary modeling to predict metagenomes from 16S data and a reference genome database [28]. 16S sequence raw data were analyzed using Qiime2 (version 2019.4). Then, the PICRUSt2 tool (version 2.1.2) was used to predict the functional profiling. As a parameter for the analysis, a cut-off equal to 2 was set to specify how distantly placed the sequence phylogeny needed to be before it was excluded. The mp (maximum parsimony) prediction method was used. The accuracy prediction is related to the presence of closely representative genomes. Lower values reveal a closer mean relationship [28]. The NSTI (Nearest Sequenced Taxon Index) test was used for assessing the accuracy of metagenome predictions. The results were analyzed using the Kyoto Encyclopedia of Genes and Genomes (KEGG) [29] and schematized via a treemap plot using the R package “Treemap Visualization” version 2.4-2.

## 3. Results

### 3.1. NMR Spectroscopy of the Compost

Most often, the assessment of the microbial activity responsible for the composting process is separated from the molecular characterization of the product resulting from this activity. Here, we chose solid-state ^13^C NMR spectroscopy to provide a direct molecular characterization of the composted biomass from which the thermophilic strains were isolated and identified using molecular techniques. Five chemical shift regions in the ^13^C-CPMAS-NMR spectrum of the resulting compost were assigned to the main organic functional groups: 0–45 ppm (aliphatic C), 45–60 ppm (O-substituted alkyl C), 60–110 ppm (O-alkyl C), 110–145 ppm (aromatic C), 145–160 (O-aryl C), and 160–190 ppm (carboxyl C) (Figure 2). Useful indicators of the hydrophobic or hydrophilic characters of the substrates were calculated from the ^13^C-CPMAS-NMR spectra as the hydrophobicity/hydrophilicity (HB/HI) ratio (Table 1).

In fact, hydrophobicity corresponds to the sum of areas under the alkyl, aryl, and O-aryl C signals, while hydrophilicity refers to the sum of the signals areas for methoxyl, O-alkyl, anomeric, and carboxyl C. The total distribution of the carbon nuclei and related structural indexes are reported in Table 1: the hydrophobicity index (HB/HI), as the ratio of the sum of hydrophobic components (aryl C + phenol C + alkyl C) over that of hydrophilic molecules (carboxyl C + O-alkyl C), and the alkyl/hydroxyalkyl index (A/OA), as the ratio of signal intensity in the 0–45 ppm range (alkyl C) over that in the 60–110 interval (O-alkyl C).

### 3.2. Isolation and the Phenotypic and Phylogenetic Characterization of Isolates Using a Culture-Dependent Approach

Strains CAF1, CAF2, and CAF5 were isolated from the enrichment media used (strains CAF2 and CAF5 were from the NB medium, while strain CAF1 was from the YN medium). All strains optimally grew aerobically at 60 °C. Strains were Gram-positive, rod-shaped, spore-forming, and oxidase- and catalase-positive. Colonies of strain CAF1 were cream and opaque with irregular margins, while strain CAF2 exhibited large colonies that were cream, smooth, and circular with a shiny surface. Strain CAF5 showed dense white growth on the agar plate. Strains CAF2 and CAF5 were negative for hydrolyses of starch, gelatine, xylan, and CMC, while strain CAF1 was positive for xylan and gelatine digestion and produced cytosolic xylanase, β-xylosidase, and α-arabinofuranosidase.

Microbial isolates were identified using the EzTaxon-e server (http://www.ezbiocloud.net/eztaxon), and based on the 16S rRNA gene sequence data, strain CAF1 was strongly related to *Geobacillus thermodenitrificans* subsp. *calidus* (100% similarity), strain CAF2 was related to *Aeribacillus pallidus* (100%), and strain CAF5 was related to *Ureibacillus terrenus* (99.71%). The 16S rRNA gene sequences of strains CAF1, CAF2, and CAF5 were deposited in GenBank/EMBL/DDBJ under the following accession numbers: MT682539, MT682538, and MT682537, respectively.

The phylogenetic tree, based on the maximum likelihood method, showed that strain CAF1 formed a clade with *Geobacillus thermodenitrificans* subsps. *calidus* and *G. thermodenitrificans*, while strain CAF2 formed a clade with *Aeribacillus pallidus* and *A. composti* (Figure 3).

### 3.3. Metagenomic Analysis: 16S rRNA Gene Amplicon Sequencing

DNA extracted from caffE showed the best quality parameters (260/280 1.8. and 260/230 1.3). Data from the 16S rRNA gene amplicon sequencing are reported in Table 2.

The total number of bacterial and archaeal reads were between 95,404 and 190,366 for the three replicates. Following the trimming step, the number of high-quality reads was 75% of the total number. Overall, the bacterial and archaeal reads were resolved in a total of 3265 (between the three replicates) and 417 OTUs, respectively. The total archaeal affiliated reads were the 1% of total community.

Relatively high alpha diversity was found for Bacteria (between 7.1 and 7.9) and the values were quite similar between replicates; in contrast, Archaea showed alpha diversity values between 1.96 and 2.18.

#### 3.3.1. Bacteria

Firmicutes dominated in the analyzed coffee compost sample (67.8% of the total sequences), followed by Proteobacteria (27.4%) and Actinobacteria (3.0%). Proteobacteria were mainly represented by Alphaproteobacteria (25.2%) and, to a lesser extent, Gammaproteobacteria (2.1%).

On average, 43.6% of the total high-quality bacterial sequences were classified at the genus level, with a total of 41 genera (0.01–15.6% of total sequences) that were detected (Table 3). A phylogenetic tree of the bacterial composition is reported in Appendix A.

The highest number of retrieved genera were affiliated with Firmicutes, with the predominant genus being *Bacillus* (5.8%), followed by *Weissella* (4.1%) and *Lactobacillus* (3.9%). Other genera, mostly affiliated with the order Bacillales, were less represented (e.g., *Rummeliibacillus*, *Lysinibacillus*, *Thermobacillus*, *Ornithinibacillus*, *Paenibacillus*, *Brevibacillus*, *Ammoniibacillus*, *Ureibacillus* and *Terribacillus* in the range of 0.1–1%). Among Alphaproteobacteria, *Acetobacter* was the best-represented genus in the coffee compost sample (15.7% of the total sequences), followed by *Ameyamaea* (5.9%). The genera *Brevundimonas*, *Shinella* and *Paracoccus* were retrieved at very low percentages. Finally, among the Actinobacteria, the dominant genus was *Saccharopolyspora* (1.8%), followed by *Saccharomonospora* and *Pseudonocardia* (both at 0.4%).

#### 3.3.2. Archaea

In general, the archaeal community was constituted by Thaumarchaeota (60%) and Euryarchaeota (15%). A total of 25% of archaeal sequences were related to unclassified Archaea (Figure 4).

Of the total high-quality archaeal sequences, about 25% were not classified at the genus level. A total of six genera were resolved from the rest (Table 4). The retrieved Thaumarchaeota almost all referred to the genus *Nitrosarchaeum* (60.9%), with 2% of them being related to *Nitrosarchaeum limnium*. Euryarchaeota were represented by a higher number of genera, mainly *Halobacterium* (13.2%) and methanogens (1.7%).

### 3.4. Functional Profiling

In general, the analysis revealed that the most represented KEGG pathway was DNA synthesis (47.7%), followed by N-acetylmuramoyl-L-alanine amidase activity (15.02%; GO:0008551), acetyl-CoA carboxylase activity (12.3%; GO:0003989), and pyruvate dehydrogenase activity (12.8%; GO:0015623). Finally, fructose-bisphosphate aldolase (8.5%; GO:0004332) and peptidyl-prolyl cis–trans isomerase activities were less observed (3.7%; GO:0003755). Results were separated by their frequency, representing the proportion of retrieved terms within the underlying protein annotation database (frequency in Db). Terms that showed a frequency higher than 1% represented more general functions, whereas a frequency lower than 0.5% was related to more specific ones. In details, more general terms were related to biological processes (among anabolic and catabolic ones) in the following order: DNA-directed DNA polymerase activity > pyruvate dehydrogenase (acetyl-transferring) activity > N-acetylmuramoyl-L-alanine amidase activity > acetyl-CoA carboxylase activity > peptidyl-prolyl cis–trans isomerase activity > fructose-bisphosphate aldolase activity. Specific terms were retrieved for the same biological processes following the same order of general terms, except for the peptidyl-prolyl cis–trans isomerase activity, which was not retrieved.

Molecular functions were completely different in general and specific terms. In the specific category, we were able to find both anabolic and catabolic molecular functions that were involved in different metabolic processes. Most of the retrieved processes were related to the cell wall and peptidoglycan degradation (in particular, phosphatidylglycerol-membrane-oligosaccharide glycerophosphotransferase activity, glycerol-3-phosphate O-acyltransferase activity, and glyceraldehyde-3-phosphate dehydrogenase (NAD^+^) (non-phosphorylating) activity), which showed an abundance of 11.1% of the total retrieved functions. Specific functionalities related to the metal ion transport, iron import into the cell, and iron-binding and amidase activity were also predicted (in particular, iron-chelate-transporting ATPase activity, cadmium-exporting ATPase activity, zinc-exporting ATPase activity, ferric-transporting ATPase activity, iron-chelate-transporting ATPase activity, zinc-exporting ATPase activity, cadmium-exporting ATPase activity, methyl indole-3-acetate esterase activity, indoleacetamide hydrolase activity, and nicotinamide-nucleotide amidase activity), representing 27.05%. The results are graphically summarized in a tree map (Figure 5).

## 4. Discussion

Composting sites represent one of the anthropocentric environments in which it is possible to find extremophiles, which are organisms that are capable of living in particular ecosystems that are characterized by extreme parameters of temperature, pH values, salinity, etc. The increased attention to environmental problems, including the aims to reduce the amount of waste and the use of non-renewable materials, as well as turning organic wastes into a valuable resource via sustainable approaches, have encouraged the use of compost in many countries [30]. Therefore, the exploitation of high-quality compost is related to many different useful applications, such as soil amendment, fertilization, and restoration.

### 4.1. Isolation and the Phenotypic and Phylogenetic Characterization of Isolates Using a Culture-Dependent Approach

Bacteria constitute the predominant group in composting processes as they possess the capability to attack highly complex organic substrates by releasing a wide range of extracellular enzymes. Thus, their characterization becomes fundamental in understanding the overall treatment process [31]. The potential use of compost as a source of thermophilic bacteria is well known [13,14,15]. In this study, we investigated the prokaryotic community in 63 °C compost (during the “early active thermophilic phase”) that was derived from waste biomass, i.e., ground coffee from an incorrect roasting process by combining culture-dependent and culture-independent methods. The Sanger sequencing of cultures allowed us to identify three diverse isolates, all of which belonging to the Bacillales order; they were three thermophilic microorganisms, growing optimally at 60 °C, namely *Geobacillus thermodenitrificans* subsp. *calidus* strain CAF1, *Aeribacillus pallidus* strain CAF2, and *Ureibacillus terrenus* strain CAF5. *Aeribacillus* and *Geobacillus* sequences were not detected by the culture-independent method applied in this study, thus further reinforcing the importance of applying different approaches to gain a more comprehensive overview of the microbial community that is present in a specific sample. Despite the fact that they did not represent particularly important community members, the isolation procedure probably allowed for their selection. Interestingly, through a partial screening of glycosyl hydrolase activities in the extracellular and intracellular comparts of isolates obtained from this work, strain CAF1 produced extracellular xylan and gelatine hydrolyses and intracellular endoxylanase, β-xylosidase, and α-arabinofuranosidase. Therefore, a deep investigation of the metabolic pathways of new isolates is desirable for confirming their suitability as cell factories for waste valorization, especially for lignocellulosic biomass, where the isolate CAF1 exhibited a multienzymatic system that was able to accomplish full hydrolyses. The relevance of those enzymes relies on the abundance of beta 1,4-xylan in nature; it results in the second most widespread polymer, representing one-third of the available renewable biomass. Xylanases are commercially important due to their ability to catalyze the bioconversion of lignocellulosic material and agro-wastes into high-value products, such as biofuels, furfural, xylitol, and artificial low-calorie sweeteners. On the other hand, although the *Geobacillus* genus is known because it consists of numerous species with various hydrolytic activities [32], for the species present in the same clade of the CAF 1 strain, no data concerning xylan degrading enzymes are reported in the literature. The discovery of novel thermophilic isolates from compost also presents an alternative to fungal-mediated lignin degradation since fungi are mostly suppressed at the thermophilic stage [33]. The opportunity of new high-throughput sequencing technologies that are nonculture-based approaches allowed us to characterize microbial communities, opening a large window on microbial diversity regarding classical culture-based approaches. Although these methods provided excellent indications of the community, it did not allow for the isolation of organisms involved in the process. For these reasons, a culture-dependent method is still necessary for the isolation of vital microorganisms present in the matrix into an axenic culture in order to identify them and to exploit their biochemical characteristics [34].

### 4.2. Metagenomic Analysis: 16S rRNA Gene Amplicon Sequencing

The isolated strains were consistent with data coming from environmental DNA analysis. Other taxa detected using the amplicon analysis were likely excluded by the culture conditions selected in this work, such as the designated pH, temperature, sodium chloride content, organic matter, oxygen availability, and absence of light. To overcome this limitation, metagenomic approaches have recently been developed to explore and access the uncultured microbial community, especially in the case of complex samples, because they furnish rapid information, do not require a preliminary culturing step, and accurately permit the classification of family and genus levels [31,35].

The composition of the prokaryotic community of compost generally depends on the starting materials and the composting procedure. However, Proteobacteria, Firmicutes, Bacteroidetes, and Actinobacteria are among the bacterial phyla that are routinely found in composting, though at different relative percentages [36]. Mesophilic organic acid-producing bacteria, such as *Lactobacillus*, *Weissella*, and *Acetobacter*, are generally abundant in the early phase of the composting process [37,38]. Conversely, in the subsequent thermophilic stage, Gram-positive bacteria, such as *Bacillus* spp. and Actinobacteria, which are often indicative of well-functioning composting processes, become dominant [39,40,41].

In this study, it was noteworthy that the relatively large abundance of the Bacillales order within the whole coffee compost bacterial community indicated a shift from the mesophilic to the thermophilic phase, as previously observed [3,42]. Members of this order are known to have genes that encode enzymes involved in the degradation of cellulose and hemicellulose, which are both coffee constituents.

Acetic acid bacteria in the Rhodospirillales order are known for their agricultural applicability [43]. In this study, *Acetobacter* and *Ameyamaea* were featured as the most abundant genera. The presence of acetic acid phylotypes in the coffee compost could be explained by the fact that those bacteria use substances that are produced by lactic acid bacteria (also well represented) as growth substrates. Thus, the high concentration of *Lactobacillus* species, together with several *Acetobacter* members, suggests that they could have been remnant sequences from the beginning of the composting process when a low pH and mesophilic temperatures occurred.

The low abundance of Actinobacteria was surprising as they generally contribute to an efficient and faster composting process during the thermophilic phase [44,45]. Interestingly, the dominant actinobacterial genus was *Saccharopolyspora*, whose species *Saccharopolyspora rectivirgula* and relatives (as well as *Saccharomonospora* spp.) are among the major contributing agents for extrinsic allergic alveolitis (also known as hypersensitivity pneumonitis) in workers of the agricultural sector, but also in personnel at composting plants [46], thus highlighting the potentially harmful impact of bioaerosols emitted by composting plants on the public’s or workers’ health [47].

Biodegradative processes in composting are supported by microbially mediated transformations of nitrogen, including ammonification (organic nitrogen in the starting materials is released as ammonium) and nitrification (oxidation of ammonium to nitrate). Since compost is widely used as a soil amendment, nitrification is particularly important as it greatly affects the availability of inorganic nitrogen forms to plants [48]. In the coffee compost analyzed in this study, the archaeal component of the prokaryotic community was dominated by ammonia-oxidizing Archaea (AOA) within Thaumarchaeota, thus leading us to suppose that ammonification (a process in which organic nitrogen in the fresh substrates can be released as ammonium) could be a prevalent process in the early thermophilic phase of the analyzed ground coffee compost. de Gannes et al. [49] reported AOA as significant contributors to nitrification in tropical compost systems. For a long time, it was thought that aerobic ammonia oxidation was carried out exclusively by chemolithoautotrophic bacteria, which are known as ammonia-oxidizing bacteria (AOB). However, this concept changed after the discovery of Thaumarchaeota, which harbors the gene for ammonia monooxygenase, being crucially involved in the ammonia oxidation process [50]. Interestingly, Thaumarchaeota sequences were mainly affiliated with the *Nitrosoarchaeum* genus, as was only recently described with the novel species *Nitrosoarchaeum koreense* from agricultural soils [51,52]. To date, few reports are available on the detection of *Nitrosoarchaeum* sequences, including those related to *N. limnium*, and they refer to floodplains [51,53]. Among Euryarchaeota, the moderate occurrence of sequences related to extremely halophilic *Halobacterium* species suggests that the analyzed coffee compost experienced a large salinity phase. Finally, the small number of methanogen sequences was surprising as thermophilic methanogens are generally ubiquitously present in compost [48].

### 4.3. Functional Profiling

Information on the functional metabolic potential of the microbiota associated with thermophilic composting is still scarce. In this study, we attempted to predict the functional composition of prokaryotes in an NMR-characterized coffee compost. The detected average sensitivity value of the metagenomic annotations (i.e., 0.11 ± 0.02 NSTI) was comparable to that reported for soil samples [22]. Results from the PICRUSt method were consistent with those reported by Wang et al. [6] and Yanan et al. [3] for the thermophilic phase, with them being characterized by a high abundance of sequences related to carbohydrate metabolism. This finding was closely related to metabolism by Bacillales, which predominated in the analyzed sample.

Interestingly, most abundant biological processes retrieved in this study showed high importance for the eventual use of coffee compost as a biofertilizer, such as the degradation of cell walls and peptidoglycan. The introduction of a microbial fertilizer that is capable of decomposing this latter molecule could lead to an increase in the turnover and release of C from a material that is relatively resistant to decomposition [54]. Moreover, many functions related to the chelation and transport of metal ions were predicted and are also very interesting in fertilizer fields. Ion movement in soil could contribute to nutrient bioavailability and aggregate formation in degraded soils [55,56].

## 5. Conclusions

In this study, the early thermophilic phase of a ground coffee composting process was explored for the molecular characterization of compost using NMR for the biodiversity and biotechnological potential of the prokaryotic (Bacteria and Archaea) community. Molecular characterization of mature compost is required in order to relate it to the biological process that produced it. Unfortunately, this is often forgotten in this type of work whereby the biological activity is seldom connected to its final product, which we tried to rectify in our study. In particular, solid-state NMR spectroscopy is the best choice for a general molecular characterization of carbon groups developed in compost without disturbing the samples with extractions, which are seldom exhaustive and often create unwanted artefacts. Among the whole bacterial community, the relatively large observed abundance of the Bacillales order (phylum Firmicutes) indicated a shift from the mesophilic to the thermophilic phase of the composting processes. This finding was also strictly related to the ability of Bacillales to degrade coffee constituents, such as cellulose and hemicellulose. In addition, the culture-dependent approach provided three thermophilic Gram-positive, spore-forming strains, and based on the 16S rRNA gene sequence similarity, the isolates were found to be Firmicutes that were strongly related to *Geobacillus thermodenitrificans* subsp. *calidus*, *Aeribacillus pallidus*, and *Ureibacillus terrenus* (strain CAF1, CAF2, and CAF5, respectively). Among them, strain CAF1 exhibited interesting enzymes, such as extracellular xylan and gelatine hydrolases and intracellular endoxylanase, β-xylosidase, and α-arabinofuranosidase. The biotechnological potential of this microorganism and its biocatalysts for the transformation and valorization of residual biomass is both large and auspicious. Therefore, deep exploration and exploitation of the strain CAF1 enzymatic pattern, which was isolated from the ground coffee compost, may promote its use in hemicellulose waste biotransformation, in line with a circular economy based on biorefinery concepts. The analysis of the Archaeal community indirectly suggested that ammonification was a prevalent process in the analyzed coffee compost (due to the occurrence of AOA within Thaumarchaeota), which probably experienced a large salinity phase (due to the occurrence of *Halobacterium* spp. within Euryarchaeota). More interestingly, coffee compost should be a novel source of *Nitrosoarchaeum* species (Thaumarchaeota), which was only recently described and to date only reported for floodplains.

Relevant biological processes (e.g., the degradation of peptidoglycan and metal chelation) detected using the predictive metabolic profiling indicated that coffee compost may be exploited as a biofertilizer for the revitalization and fertilization of agricultural soils by increasing, for instance, the turnover and release of carbon, as well as the nutrient bioavailability.

Thanks to the combination of culture-dependent and -independent methods, our findings underline that organic waste, such as burnt ground coffee, possessed a very interesting microbiological profile. Coffee compost could be applied as a biofertilizer in the agricultural fields and used for the isolation of biotechnologically relevant bacteria.

## Figures and Tables

**Figure 1 microorganisms-09-00218-f001:**
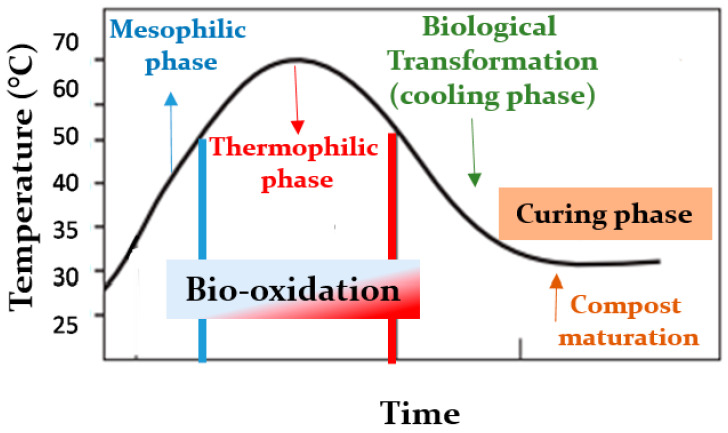
Temperature changes in the composting process.

**Figure 2 microorganisms-09-00218-f002:**
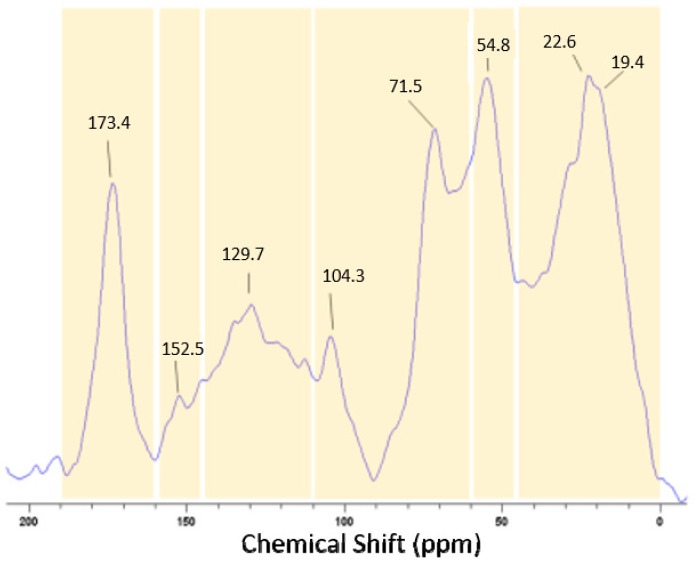
^13^C-CPMAS-NMR spectrum of green ground coffee compost; the numbers above the peaks represent the chemical shift signals assigned to the main organic functional groups: 0–45 ppm (aliphatic C), 45–60 ppm (O-substituted alkyl C), 60–110 ppm (O-alkyl C), 110–145 ppm (aromatic C), 145–160 (O-aryl C), and 160–190 ppm (carboxyl C).

**Figure 3 microorganisms-09-00218-f003:**
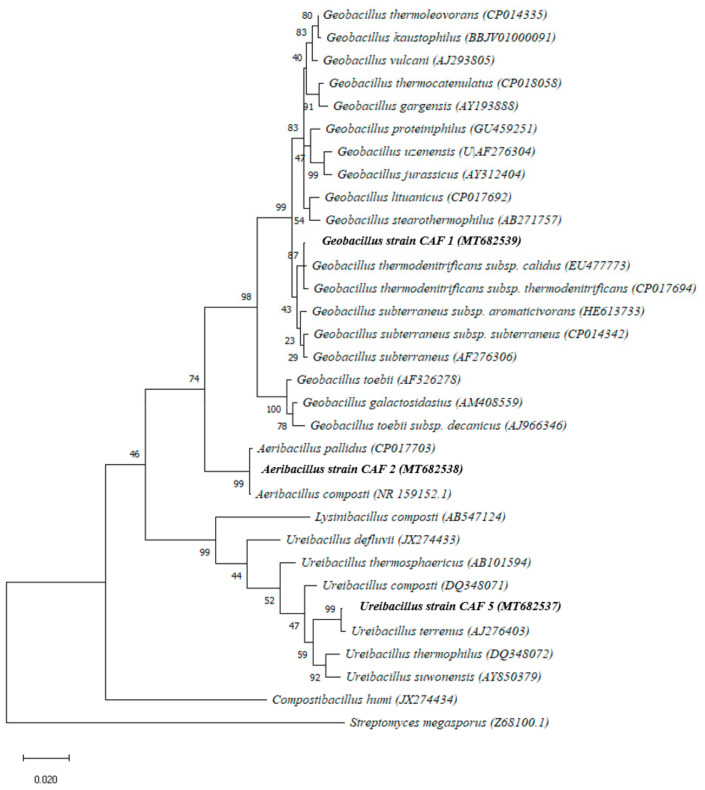
Evolutionary phylogenetic tree that was constructed using the maximum likelihood method showing the relationship between the strains CAF1, CAF2, and CAF5 and the related taxa. Bootstrap values based on 1000 replications are shown at nodes. The isolates obtained from this work are reported in bold. Numbers *Streptomyces megasporus* INA M-22T (Z68100) was used as an outgroup.

**Figure 4 microorganisms-09-00218-f004:**
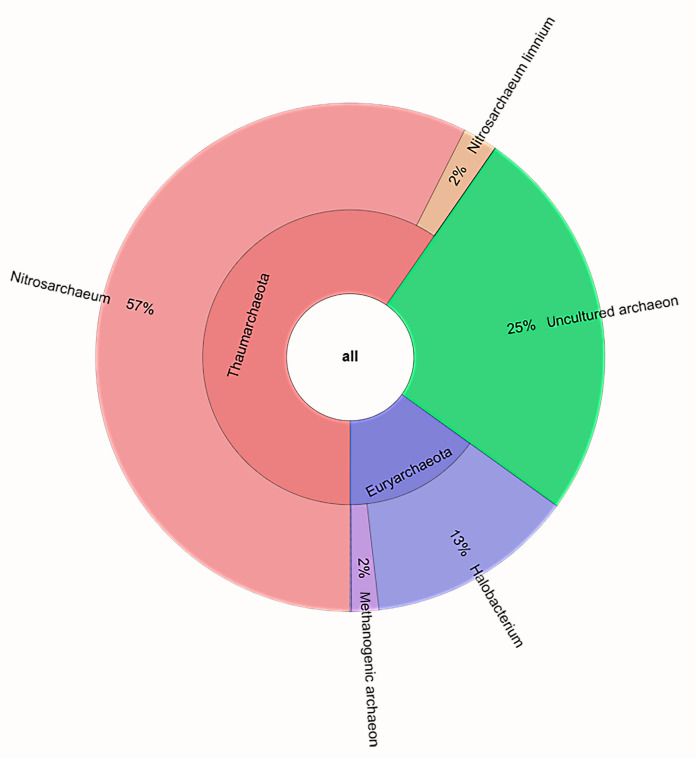
Archaeal phyla and genera. Graph generated using the krona tool.

**Figure 5 microorganisms-09-00218-f005:**
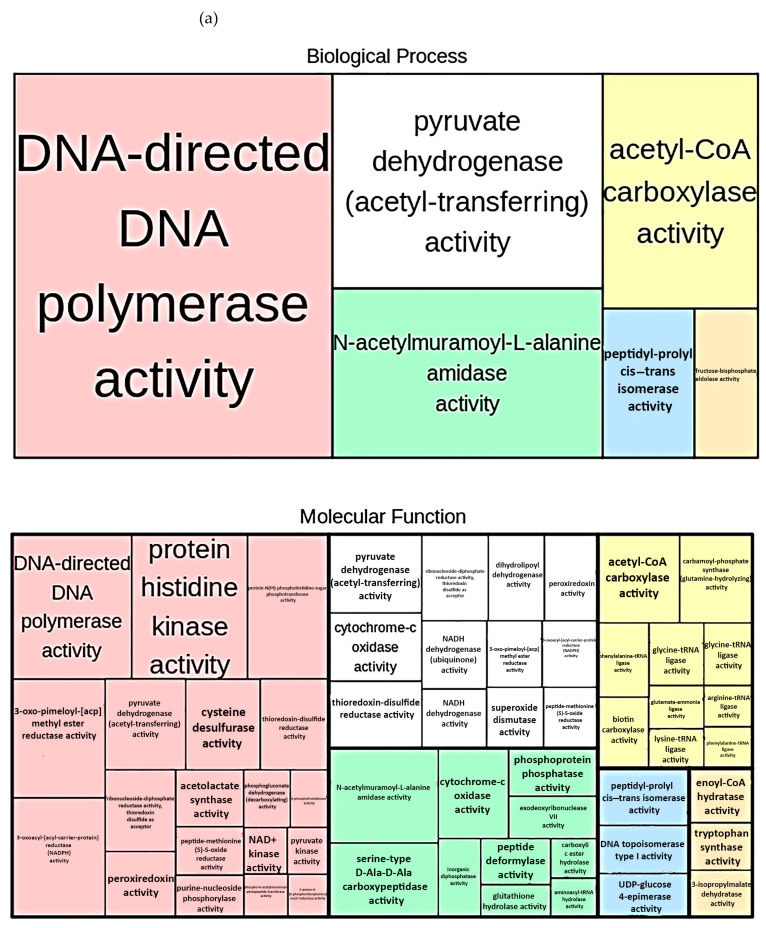
Tree map of biological processes and molecular functions as predicted using the PICRUSt2 software and analyzed against the KEGG database: (**a**) molecular functions and the biological processes in which they were retrieved, showing a frequency of Db > 1%, and (**b**) molecular functions and the biological processes in which they were retrieved, showing a frequency of Db < 0.5%.

**Table 1 microorganisms-09-00218-t001:** Relative carbon percent distribution over different chemical shift (ppm) intervals of the cross-polarization magic-angle spinning–nuclear magnetic resonance (CPMAS-NMR) spectrum of the compost and the corresponding indexes.

Carbon Species (ppm Intervals)	Aliphatic C (0–45)	O-Substituted Alkyl C (45–60)	O-Alkyl C (60–110)	Aromatic C (110–145)	O-Aryl C (145–160)	Carboxyl C (160–190)	HB/HI ^a^	A/OA ^b^
Relative %	33.5	15.2	23.3	14.8	3.2	10.0	2.7	0.7

^a^ Hydrophobicity index HB/HI = ((0–45) + (145–160))/((45–60) + (160–190)) and ^b^ alkyl/hydroxyalkyl index A/OA = (0–45)/(45–60), where (0–45) refers to the signal intensity in the 0–45 ppm range, and similarly for the other ranges.

**Table 2 microorganisms-09-00218-t002:** Results of the next-generation sequencing, the number of good quality reads, and operational taxonomic units (OTUs) obtained from the data analysis. The retrieved diversity is associated with replicates of the sample.

Reads						Bacteria OTUs		Archaea OTUs	
DNA Replicate Name’s Samples	Total Number	Clipping- Trimming- Filtering	Good Quality (%)	Reads Missed (%)	Final Reads Number	Number	Shannon Index	Number	Shannon Index
caffA	95,404	71,098	74.52	25.47	65,837	1439	7.12	99	1.996603
caffB	126,649	91,468	72.22	27.77	85,614	2183	7.46	164	1.969011
caffE	190,366	142,988	75.11	24.88	133,122	3265	7.95	253	2.184408

**Table 3 microorganisms-09-00218-t003:** Genera retrieved within the bacterial community.

Phylum/Class	Genus	Proportion of Total Reads Number (%)				
		caffA	caffB	caffE	Average	St.Dev.
Actinobacteria	*Frigoribacterium*	0.00	0.00	0.03	0.01	±0.02
	*Sanguibacter*	0.07	0.00	0.00	0.02	±0.04
	*Pseudonocardia*	0.28	0.37	0.58	0.41	±0.16
	*Saccharomonospora*	0.14	0.27	0.74	0.38	±0.31
	*Saccharopolyspora*	0.35	0.64	4.38	1.79	±2.25
	*Streptomyces*	0.07	0.00	0.06	0.04	±0.04
	*Nocardiopsi*	0.00	0.05	0.03	0.03	±0.02
	*Thermomonospora*	0.00	0.05	0.00	0.02	±0.03
Bacteroidetes	*Persicitalea*	0.00	0.05	0.03	0.03	±0.02
	*Parapedobacter*	0.00	0.14	0.00	0.05	±0.08
Firmicutes	*Aeribacillus*	0.00	0.05	0.09	0.05	±0.05
	*Bacillus*	7.71	5.54	4.26	5.84	±1.75
	*Oceanobacillus*	0.21	0.78	0.06	0.35	±0.38
	*Ornithinibacillus*	0.28	0.14	0.34	0.25	±0.10
	*Terribacillus*	0.07	0.00	0.00	0.02	±0.04
	*Ureibacillus*	0.97	0.87	1.01	0.95	±0.07
	*Ammoniibacillus*	0.14	0.05	0.03	0.07	±0.06
	*Brevibacillu*	0.21	0.00	0.06	0.09	±0.11
	*Paenibacillus*	0.14	0.23	0.28	0.21	±0.07
	*Thermobacillus*	0.00	0.00	0.15	0.05	±0.09
	*Bhargavae*	0.00	0.14	0.00	0.05	±0.08
	*Lysinibacillus*	1.74	1.33	0.98	1.35	±0.38
	*Rummeliibacillus*	0.00	0.00	0.06	0.02	±0.04
	*Lactobacillus*	2.43	3.34	6.09	3.96	±1.91
	*Pediococcus*	0.07	0.05	0.28	0.13	±0.13
	*Leuconostoc*	0.00	0.00	0.03	0.01	±0.02
	*Weissella*	2.15	3.34	6.89	4.13	±2.46
	*Lactococcus*	0.00	0.00	0.03	0.01	±0.02
	*Tepidimicrobium*	0.00	0.00	0.15	0.05	±0.09
Alpha	*Acetobacter*	17.16	21.48	8.33	15.66	±6.70
	*Ameyamaea*	7.30	6.46	3.86	5.87	±1.79
	*Brevundimonas*	0.07	0.00	0.00	0.02	±0.04
	*Shinella*	0.00	0.00	0.03	0.01	±0.02
	*Paracoccus*	0.07	0.00	0.00	0.02	±0.04
	*Ameyamaea*	0.00	0.41	0.06	0.16	±0.22
Gamma	*Bordetella*	0.83	0.14	0.06	0.34	±0.43
	*Lautropia*	0.00	0.05	0.00	0.02	±0.03
	*Orrella*	0.00	0.05	0.00	0.02	±0.03
	*Kingella*	0.00	0.05	0.00	0.02	±0.03
	*Franconibacter*	0.00	0.00	0.03	0.01	±0.02
	*Alcanivorax*	0.07	0.00	0.00	0.02	±0.04
	*Pseudomonas*	1.95	0.87	0.40	1.07	±0.79

**Table 4 microorganisms-09-00218-t004:** Genera retrieved within the archaeal community.

Phylum/Class		Proportion of Total Reads Number (%)				
	Genus	caffA	caffB	caffE	Average	St.Dev.
Euryarchaeota	*Halobacterium*	12.89	13.24	13.46	13.20	±0.28
	*Halogeometricum*	0.00	0.00	0.12	0.04	±0.06
	*Methanogenic archaeon*	1.27	0.92	2.76	1.65	±0.97
	*Mathanomicrobiales archaeon*	0.00	0.00	0.12	0.04	±0.06
Thaumarchaeota	*Nitrosarchaeum*	65.53	66.93	43.50	58.65	±13.14
	*Nitrosarchaeum limnium*	2.53	2.53	1.73	2.26	±0.46

## Data Availability

The data presented in this study are available in article.

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
