# Peer review of "Prokaryotic Diversity of the Composting Thermophilic Phase: The Case of Ground Coffee Compost"

_microorganisms, 2021, doi:10.3390/microorganisms9020218_

Round 1

Reviewer 1 Report

The work carried out is of enormous interest to experts in composting microbiology. An immense amount of work can be guessed, which has resulted in very valuable information. In addition, the great diversity of applied techniques, basic and advanced, represent an added value to the work carried out. However, despite being an almost excellent work, the objective of the research is confused when techniques of both a cultivable and molecular nature are approached. Furthermore, it is not clear whether what is being pursued is a study of biodiversity or simply the isolation of strains with some biotechnological application.

Here are some recommendations that could improve the final quality of the work.

1- The authors must better define the objectives of the work both in the “Abstract” and in the “Introduction” (is it a study of the biodiversity of the thermophilic phase of the composting process?, Is it a study aimed at isolating strains with biotechnological potential?, Is it a predictive study? ...)

2- Line 57, I don't totally agree with this sentence: “However, Knowledge of the microbiology of this process is limited…”. Please, see the following references:

Estrella-González, M.J. Suárez-Estrella, F. Jurado, M.M. López, M.J. López-González, J.A. Siles-Castellano, A.B. Muñoz-Mérida, A. Moreno, J. (2020). Uncovering new indicators to predict stability, maturity and biodiversity of compost on an industrial scale. Bioresource Technology. 313,123557.

Jurado, M.M. Camelo-Castillo, A.J. Suárez-Estrella, F. López, M.J. López-González, J.A. Estrella-González, M.J. Siles-Castellano, A.B. Moreno, J. (2020). Integral approach using bacterial microbiome to stabilize municipal solid waste. Journal of Environmental Management 265,110528.

López-González, J.A., Suárez-Estrella, F., Vargas-García, M.C., López, M.J., Jurado, M.M., Moreno, J. (2015). Dynamics of bacterial microbiota during lignocellusic waste composting: Studies upon its structure, functionality and biodiversity. Bioresource Technology 175, 406-416.

López-González, J.A., Vargas-García, M.C., López, M.J., Suárez-Estrella, F., Jurado, M.M., Moreno, J. (2015). Biodiversity and succession of mycobiota associated to agricultural lignocellulosic waste-based composting. Bioresource Technology 187, 305-313.

Jurado, M. M., Suárez-Estrella, F., Vargas-García, M. C., López, M. J., López-González, J. A., Moreno, J. (2014). Increasing native microbiota in lignocellulosic waste composting: Effects on process efficiency and final product maturity. Process Biochemistry 49(11), 1958-1969.

Jurado, M.M., López, M. J., Suárez-Estrella, F., Vargas-García, M. C., López-González, J. A., Moreno, J. (2014). Exploiting composting biodiversity: Study of the persistent and biotechnologically relevant microorganisms from lignocellulose-based composting. Bioresource Technology 162, 283-293.

López-González, J.A., Vargas-García, M.C., López, M.J., Suárez-Estrella, F., Jurado, M.M., Moreno, J. (2014). Enzymatic characterization of microbial isolates from lignocellulose waste composting: chronological evolution. Journal of Environmental Management 145, 137-146.

2 - Justify the need to use NMR spectroscopy techniques in this work

3 - There is an excessive number of references (more than 60). Some of them are quite old (more than 20 years ago). Use the name of the journal according to the authors' guide (long or short name format, but not both).

4 - Figures 3 and 4 show very low quality. The explanation of Figure 4 in the text is not sufficient (Figures 4a and 4b do not differ in subsection 3.4).

5 – Like the objectives of the work, the conclusions are confusing. In this case, the importance of the isolated strains, their biotechnological potential and their possibilities of commercial exploitation are highlighted. However, the enormous relevance of the study of microbial biodiversity of the thermophilic phase of composting, as well as the predictive nature of the work, goes unnoticed. I recommend rewriting the conclusions of the work.

Reviewer 2 Report

  The study by Papale et al. investigates the community composition of coffee-compost samples at the thermophilic phase of composting by 16S gene amplicon sequencing and cultivation attempts. The bacterial and archaeal community is described, and a functional community profile is predicted based on previously characterized relatives by PICRUSt. Three novel isolates are presented, and their ability to degrade complex polysaccharides is examined. If the aim of this study is to convince readers that coffee compost is a great source of microbial guilds capable of sustainable organic matter decomposition and fertilization, it fails to do that because it remains largely descriptive and the most interesting findings are scarcely analysed (the functional potential of the community), or compared to the existing literature. For example, has there been a previous attempt to characterize coffee-compost communities? Or in general how do the functional profiles compare to other types of compost?

The biological processes identified should be discussed in the context of the microbial guilds present, in addition to the processes inferred by the present microbes themselves, in order to come up with a model for the dominant cycles taking place (carbon and nitrogen). For example, which groups do the authors propose mediate the major part of carbon cycling in this sample? 

Also, while it is interesting to compare the cultivation results with the metagenomic community profiling to assess cultivation biases, the isolated microbes do not represent dominant community members. What is then the argument for their significance? Are the functional profiles unique?  

The manuscript would benefit from english proofreading, as the syntax in certain places is sometimes confusing.  

Specific comments:  

lines 51-56: A graph of the different phases of composting would be helpful for readers not so familiar with the process.   

Phylogenetic analysis: You mention that the trees were reconstructed by using MEGA X with the neighbour-joining method. In general, the most robust and commonly used methods are maximum likelihood and Bayesian methods which can employ sequence evolution models, since NJ is just a clustering algorithm. MEGA 7 has a composite model option that uses the NJ method to estimate distance and tree topology and ML to estimate the support of the tree, but this was not used in the article. I would ask to reconstruct the phylogenetic trees by using appropriate ML methods, or the composite option from MEGA 7.   

line 157: Please give the names of the primers also.   

paragraph 205-213: please correct the syntax and grammar. Also, there is a new PICRUSt2 paper: Douglas, G.M., Maffei, V.J., Zaneveld, J.R. et al. PICRUSt2 for prediction of metagenome functions. Nat Biotechnol 38, 685–688 (2020). https://doi.org/10.1038/s41587-020-0548-6  

Section 3.1: These results are not discussed at all further. What do they contribute to our understanding of the community and processes? Even if just descriptive, they need to be integrated with the rest of the results, otherwise I do not know what we are meant to make of this.   

Table 2: If caffA, B & E are just DNA extraction replicates, why is the number of reads obtained from caffE almost double? Were there any parameters changed between extractions? This results in a much higher number of OTUs for both Bacteria and Archaea, so it has an impact on the analysis.   

The isolated bacteria do not belong to the most abundant community members in the compost samples. While the limitations of culture-based methods are well-known, an explanation would need to be provided as to why that is.  Also, the authors would then need to argue about  the significance of these isolates, since they do not represent particularly important community members.   

paragraph 3.4: This section is not understandable. If these are the PICRUSt results, based on what marker genes are these pathways detected? Are they catabolic or anabolic?

lines 309-312 are very unclear: please describe more clearly what exactly you mean by frequency of the terms in the annotation databases, and what does this mean in terms of the community.

Figure 4 is also hard to understand: how can the same process (e.g. DNA directed DNA polymerase activity) have both >1% and <0.5% frequency in the database?? Also, given the “nested” appearance and color-coding of the boxes in  Fig.4, are we meant to assume that all the “Biological process” boxes contain all the “molecular function” boxes respectively? If this is the case, then I do not see for example how glutamate synthase activity is contained in the DNA-directed DNA polymerase biological process. Please rewrite this section.   

The discussion paragraph needs to be organised into sub-sections for clarity. 

lines 340-350: Better comparison of the novel isolates to previously described strains. Are the activities observed for CAF1 a novelty in this clade or are they common activities found in Geobacillus?  

lines 381-387: Lactobacillus represent a 3.9% of the compost community, while Acetobacter and Ameyamaea together almost 20%, both much higher than the supposedly active genera in the thermophilic stage. Is it really possible that they are just remnant sequences from the mesophilic stage? Or could there be another explanation?  

line 390: Saccharopolyspora is overrepresented in just one of the three replicates (caffE). Could this be an artefact/ overinterpretation of a spurious pattern?   

line 401: If you mean that ammonification is a prevalent process providing adequate substrate and therefore explaining the high abundance of Thaumarchaeota, please clarify. 

line 407: N. koreense was isolated from agricultural soil.   

paragraph 413-422: The purported "biological processes of great importance” are only superficially mentioned and scarcely analysed.   

line 421: The last sentence sounds pretty random here. What is the logical step from the previous sentence, which talks about degradation of recalcitrant carbon?  

lines 435-443: This belongs to introduction or discussion. It is not the right place to introduce the significance of xylans. Also the conclusions need to be rewritten to highlight the main messages of this study, not just describe which genera were found.     

Round 2

Reviewer 1 Report

I agree with the modifications and improvements made by the authors.

Author Response

Manuscript ID1025179

Papale et al. “Prokaryotic Diversity of the Composting Thermophilic Phase: the Case of Ground Coffee-Compost”.

Dear Editor,

Thank you for your response and reviewer comments on our manuscript ID1025179.

We have revised the manuscript based on the peer and expert comments offered by the reviewers. The enclosed document lists our point-by-point response to the reviewers’ comments.

Hoping to have fully addressed the reviewers’ comments, please consider the revised version of manuscript.

Yours sincerely,

Dr. Annarita POLI, on behalf of the authors

Response to reviewers’ comments

REVIEWER 1

English language and style

English language and style are fine/minor spell check required

  1. We have checked English language and corrected minor errors.

Reviewer n.2

English language and style are fine/minor spell check required

  1. We have checked English language and corrected minor errors.

Comments and Suggestions for Authors

The authors addressed most of the comments adequately, and the presentation of study has improved significantly. There are still a couple of minor points that I would like to comment on:

While the authors elaborated on the NMR analysis, it is still unclear what the reader is meant to understand from this. In other words: What does it mean that there are XX% of the main organic functional groups and why should we care? For example, does this mean the compost contains a low concentration of toxic compounds? Is there still a lot of recalcitrant carbon? Please add a clarifying sentence for the non-specialist reader.

  1. A molecular characterization of mature compost is required in order to relate it to the biological process that has produced it. Unfortunately, this is often forgotten in this type of works whereby the biological activity is seldom connected to its final product, as conversely we tried to do in our study. In particular, solid-state NMR spectroscopy is the best choice for a general molecular characterization of carbon groups developed in compost, without disturbing the samples with extractions which are seldom exhaustive and often crate unwanted artifacts.

The different results obtained from the three replicates are still somewhat puzzling. While of course we expect a degree of variability in biological replicates, we are also doing them to assess potential weak points/errors in the analysis or sampling, and find outliers. In this case, caffE is clearly an outlier: the final reads number is almost the sum of the other two. If you were to perform any statistics on this, this is beyond the range of simple standard deviation. I'm not saying to discard this sample, i'm simply suggesting that a reason for this would be needed. Was this sample taken from a different location/depth in the pile? Was there a difference in the extraction protocol? Did you sequence deeper for some reason?

R. We checked again all data trying to find some differences between samples. The only one was that caffE mixture was darker and more compact than the other samples.  DNA was extracted from the same amount of sample, and the same protocol was used. Though caffE had quite the same amount of DNA quantity of other replicates, it showed the best quality targets in terms of 260/280 and 260/230 ratio. This means that in terms of residual proteins as well as carbohydrates and phenols, the extraction of caffE had the best characteristics. Probably this could be the reason of encountered differences.

Reviewer 2 Report

The authors addressed most of the comments adequately, and the presentation of study has improved significantly. There are still a couple of minor points that I would like to comment on:

  • While the authors elaborated on the NMR analysis, it is still unclear what the reader is meant to understand from this. In other words: What does it mean that there are XX% of the main organic functional groups and why should we care? For example, does this mean the compost contains a low concentration of toxic compounds? Is there still a lot of recalcitrant carbon? Please add a clarifying sentence for the non-specialist reader.
  • The different results obtained from the three replicates are still somewhat puzzling. While of course we expect a degree of variability in biological replicates, we are also doing them to assess potential weak points/errors in the analysis or sampling, and find outliers. In this case, caffE is clearly an outlier: the final reads number is almost the sum of the other two. If you were to perform any statistics on this, this is beyond the range of simple standard deviation. I'm not saying to discard this sample, i'm simply suggesting that a reason for this would be needed. Was this sample taken from a different location/depth in the pile? Was there a difference in the extraction protocol? Did you sequence deeper for some reason?

Author Response

Manuscript ID1025179

Papale et al. “Prokaryotic Diversity of the Composting Thermophilic Phase: the Case of Ground Coffee-Compost”.

Dear Editor,

Thank you for your response and reviewer comments on our manuscript ID1025179.

We have revised the manuscript based on the peer and expert comments offered by the reviewers. The enclosed document lists our point-by-point response to the reviewers’ comments.

Hoping to have fully addressed the reviewers’ comments, please consider the revised version of manuscript.

Yours sincerely,

Dr. Annarita POLI, on behalf of the authors

Response to reviewers’ comments

REVIEWER 1

English language and style

English language and style are fine/minor spell check required

  1. We have checked English language and corrected minor errors.

Reviewer n.2

English language and style are fine/minor spell check required

  1. We have checked English language and corrected minor errors.

Comments and Suggestions for Authors

The authors addressed most of the comments adequately, and the presentation of study has improved significantly. There are still a couple of minor points that I would like to comment on:

While the authors elaborated on the NMR analysis, it is still unclear what the reader is meant to understand from this. In other words: What does it mean that there are XX% of the main organic functional groups and why should we care? For example, does this mean the compost contains a low concentration of toxic compounds? Is there still a lot of recalcitrant carbon? Please add a clarifying sentence for the non-specialist reader.

  1. A molecular characterization of mature compost is required in order to relate it to the biological process that has produced it. Unfortunately, this is often forgotten in this type of works whereby the biological activity is seldom connected to its final product, as conversely we tried to do in our study. In particular, solid-state NMR spectroscopy is the best choice for a general molecular characterization of carbon groups developed in compost, without disturbing the samples with extractions which are seldom exhaustive and often crate unwanted artifacts.

The different results obtained from the three replicates are still somewhat puzzling. While of course we expect a degree of variability in biological replicates, we are also doing them to assess potential weak points/errors in the analysis or sampling, and find outliers. In this case, caffE is clearly an outlier: the final reads number is almost the sum of the other two. If you were to perform any statistics on this, this is beyond the range of simple standard deviation. I'm not saying to discard this sample, i'm simply suggesting that a reason for this would be needed. Was this sample taken from a different location/depth in the pile? Was there a difference in the extraction protocol? Did you sequence deeper for some reason?

  1. We checked again all data trying to find some differences between samples. The only one was that caffE mixture was darker and more compact than the other samples. DNA was extracted from the same amount of sample, and the same protocol was used. Though caffE had quite the same amount of DNA quantity of other replicates, it showed the best quality targets in terms of 260/280 and 260/230 ratio. This means that in terms of residual proteins as well as carbohydrates and phenols, the extraction of caffE had the best characteristics. Probably this could be the reason of encountered differences.